# A Bayesian Study of the Dynamic Effect of Comorbidities on Hospital Outcomes of Care for Congestive Heart Failure Patients

**Dimitrios Zikos** [1,*]**, Stelios Zimeras** [2] **and Neli Ragina** [3]

1   College of Health Professions, Central Michigan University, Mount Pleasant, MI 48859, USA
2   Department of Statistics and Actuarial-Financial Mathematics, University of the Aegean,
    83200 Karlovassi, Greece; zimste@aegean.gr
3   College of Medicine, Central Michigan University, Mt. Pleasant, MI 48859, USA; ragin1n@cmich.edu
*   Correspondence: zikos1d@cmich.edu

**Abstract:** Comorbidities can have a cumulative effect on hospital outcomes of care, such as the length of stay (LOS), and hospital mortality. This study examines patients hospitalized with congestive heart failure (CHF), a life-threatening condition, which, when it coexists with a burdened disease profile, the risk for negative hospital outcomes increases. Since coexisting conditions co-interact, with a variable effect on outcomes, clinicians should be able to recognize these joint effects. In order to study CHF comorbidities, we used medical claims data from the Centers for Medicare and Medicaid Services (CMS). After extracting the most frequent cluster of CHF comorbidities, we: (i) Calculated, step-by-step, the conditional probabilities for each disease combination inside this cluster; (ii) estimated the cumulative effect of each comorbidity combination on the LOS and hospital mortality; and (iii) constructed (a) Bayesian, scenario-based graphs, and (b) Bayes-networks to visualize results. Results show that, for CHF patients, different comorbidity constructs have a variable effect on the LOS and hospital mortality. Therefore, dynamic comorbidity risk assessment methods should be implemented for informed clinical decision making in an ongoing effort for quality of care improvements.

**Keywords:** comorbidities; congestive heart failure; health informatics; Bayes networks; clustering; risk assessment; clinical decision making

---

## 1. Introduction

In recent years there is a high surge in the prevalence of chronic diseases with almost half adults having two or more chronic conditions [1]. Chronic diseases necessitate not only medical care and extensive use of healthcare resources, but also limit daily life activities and patient independence [2]. When a patient has a condition, chronic or not, in addition to his/her primary diagnosis ($D_x$), this disease is known as a comorbidity. Comorbidity is the presence of "a distinct additional clinical entity" [3] and is not related to the index condition or a side effect or result of treatment [4]. Comorbidities may or may not be directly related to the primary diagnosis [5]. For instance, a patient with a primary diagnosis of congestive heart failure (CHF) may have hypertension (related condition) and chronic psoriasis (unrelated condition). Comorbidities can lead to patient complications and may trigger the need for hospitalization, oftentimes with a high risk for increased length of stay (LOS), hospital-acquired conditions, and hospital death [6–8]. Their existence is important to be taken into account by physicians, because of their impact on the diagnosis, treatment, prognosis, and outcome [3].

Since 10,000 people in the United States turn 65 every day, there is an ongoing increase in the number of individuals with comorbidities [9]. Several studies confirmed that chronic conditions

appear together in clusters, as seen in the case of cardiovascular diseases [10–13], while some clusters of comorbidities have been shown to have a synergistic effect [14]. By identifying the aforementioned clustered clinical homogenous patient subgroups, it can become possible to develop better targeted and personalized interventions [2]. Treating patients for their comorbidity composition, and not separately for each diagnosis in a silo, may contribute to tackling health problems more effectively, with improved coordination of care and integration of practice [9], in line with a holistic and patient-centered care approach. Clinical guidelines hardly address comorbidities, and this can result in adverse events [10]. This emphasizes the importance of having patient-focused management and an efficient user-defined comorbidity system to identify risk factors and outcomes.

Researchers have studied outcomes and prognosis in patients with multiple chronic diseases. Age and at least two comorbidities were found to be strong predictors for the development of hospital-acquired conditions (HAC), which in turn impact LOS negatively [15]. Nobili et al. found that the presence of comorbidity increases mortality risk and LOS [16]. In addition, according to Parappil [17], patients with chronic obstructive pulmonary disease and comorbid diabetes have increased LOS and elevated risk of death. Prolonged LOS, in turn, has been shown to be associated with adverse outcomes [18]. Because of all the aforementioned patient safety implications, it is critical for clinical decision-makers to take into consideration the synergistic effect of various comorbidity profiles.

Using large healthcare datasets to stratify chronic conditions can contribute to improvements to the quality of care [9]. Health analytics methods, for instance, can be applied to study associations between risk factors and chronic conditions [19]. A systematic understanding of interactions between comorbidities can become possible with the support of data-driven technologies. These technologies can contribute to understanding and then planning to reduce preventable adverse outcomes of care.

In this study, we examine the effect of comorbidity constructs on the LOS and hospital mortality, among elderly patients admitted to the hospital with a primary diagnosis of CHF, in the United States. Annually, over a million admissions occur in the United States for CHF [20] with more than 6.5 million hospital days and an economic burden of $37.2 billion [21]. Elderly with CHF have at least five comorbidities and high post-discharge mortality and readmission rates [20–23]. When a primary $D_x$ of CHF is accompanied by comorbidities, patient management and treatment increases in complexity: More than 30% of patients with CHF have comorbidities that worsen during hospitalization. These patients present increased mortality risk [21,24] and prolonged hospital LOS, among other adverse hospital outcomes of care. For CHF patients, renal failure, COPD, diabetes, depression [22], hypertension, myocardial infarction, coronary artery disease, atrial fibrillation, anemia, chronic liver disease, and sleep-disordered breathing [25–27] were found to be predictors of LOS and hospital mortality.

While studies, such as the aforementioned ones, examine CHF comorbidities and their effect on hospital outcomes, they were not designed to compare outcomes of care between different comorbidity combinations. Also, existing studies do not examine the dynamic effect of a disease on outcomes, when it is added onto a pre-existing disease construct. The objective of this work is to study the cumulative effect of comorbidities on the LOS and hospital mortality, for patients who have been diagnosed with CHF. Firstly, we used partitional clustering to find the most frequent hospital comorbidity construct (cluster) for patients with a primary diagnosis of CHF. We then extracted this cluster and calculated cumulative conditional probabilities for all comorbidity combinations within this cluster. These calculations served as the foundation for a visualized collection of directed acyclic graphs and Bayes Networks that can be navigated to examine the cumulative effect of any CHF comorbidity combination, on the two outcomes under study.

## 2. Materials and Methods

### 2.1. Data Sources

This a cross-sectional study that was conducted with the use of a large medical claims dataset from CMS that contains more than 564,875 records of actual inpatient Medicare cases. The dataset was purchased after signing a Data Use Agreement with CMS. Apparently one of the advantages of Bayes methods that this study uses is that they do not require enormous amounts of data points to model associations, and with this dataset, more than 25,000 cases with a diagnosis of CHF were later extracted and included for analysis. The file contains information about patient admission, patient demographics, the International Classification of Diseases 9th edition (ICD-9-CM) diagnoses (Primary and Secondary), hospital-acquired conditions, hospital procedures, disposition information, hospital charges, and service utilization variables. Medicare datasets have been used in a variety of studies to determine patient needs, suggest required services, and understand factors associated with negative outcomes. A variety of similar large healthcare datasets have also been used to study the effect of clinical practice on health outcomes [28], compare hospitals and their patient safety performance, and allow the in-depth study of rare conditions [29].

### 2.2. Data Selection and Preparation

Since we study the comorbidities of patients hospitalized with a primary diagnosis of CHF, we firstly filtered the data, thereafter keeping only cases with a primary diagnosis of CHF. This is the target dataset that was used for the analytical phase and contained 25,647 records of patients admitted to the hospital, due to CHF (Figure 1, Data Selection Phase).

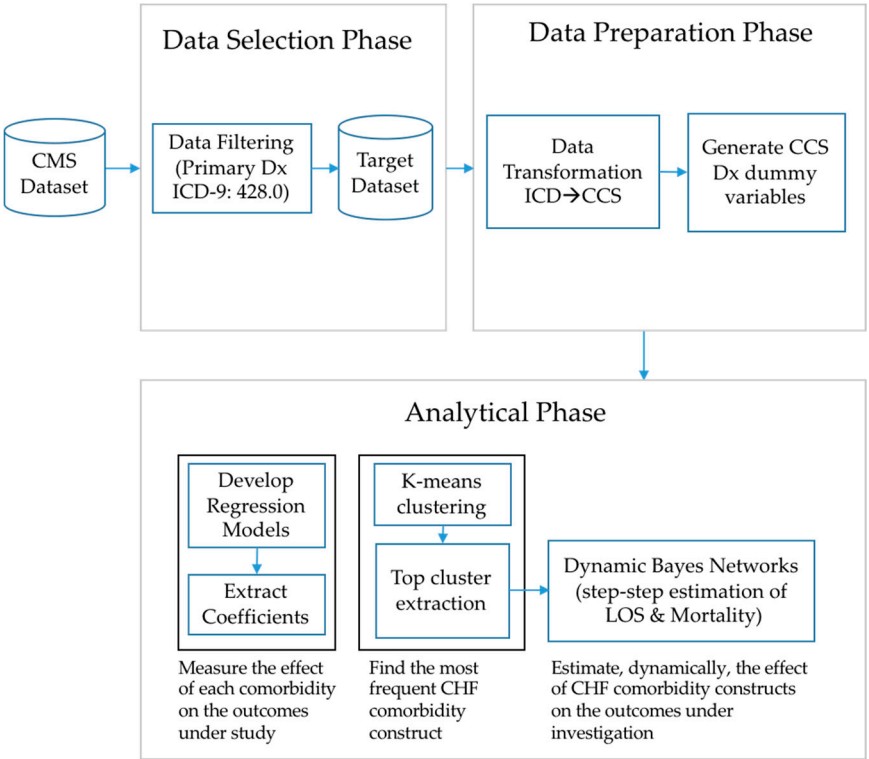

**Figure 1.** Overview of the study methodology.

The ICD codes (CHF comorbidities) were then grouped into Clinical Classification Software (CCS) codes. CCS is a grouping developed by the Agency of Healthcare Research and Quality (AHRQ) [30] to provide a meaningful clinical representation of diagnosis codes. CCS groups thousands of unique ICD codes into 285 exclusive clinical categories. To transform diagnosis attributes from the ICD to the CCS

coding system, a crosswalk, available at the AHRQ website was used. With the diagnosis attribute transformation to CCS, the data dimensionality was significantly reduced (285 dummy variables, one for each CCS code, instead of thousands of dummy ICD-9-CM variables) and the risk data overfitting was significantly minimized, with one small trade-off being small loss in diagnosis specificity (Figure 1, Data Preparation Phase).

*2.3. Analytical Phase*

The analytical phase includes three tasks (Figure 1, Analytical Phase). Firstly, a coefficient analysis was completed using regression methods described here: The LOS was examined using Multiple Linear Regression since it is a continuous variable. The study of hospital mortality, which is a dichotomous response, was completed with Binary Logistic Regression. For both regressions, independent variables were inserted into the models using the Enter method. Regression coefficients were extracted from each model to find the CHF comorbidities associated with prolonged hospital day and high rates of mortality.

The second task is the step-by-step calculation of the cumulative effect of CHF comorbidities on the LOS and hospital mortality. We decided to study contextually relevant CHF comorbidities and not just any random combination of comorbidities. For this reason, we extracted the most frequent CHF comorbidity set, by employing partitional clustering.

The third task uses this frequent cluster of CHF comorbidities. For this cluster, we estimated, in a modular manner, for every combination of comorbidities, the mean LOS and hospital mortality, step by step, and visualized the constructs using directed acyclic graphs and Bayes Networks. Figure 1 shows an overview of the study methodology.

All tasks (regression analysis, clustering, and Bayes networks) were completed with the use of Weka Machine Learning Software (University of Waikato, Hamilton, New Zealand), v.3.8.3, Windows version (https://www.cs.waikato.ac.nz/ml/weka/). Weka has been used in numerous biomedical and clinical research studies. The software is open-source and consolidates the majority of machine learning algorithms.

## 3. Results

*3.1. Data Description*

We hereby present the descriptive statistics of the data under study, for the two outcomes of LOS and mortality and the five comorbidities of Cluster 1. Table 1 shows descriptives and ranges for the entire dataset (left), and for target dataset that only includes cases with a primary diagnosis of CHF (right).

**Table 1.** Descriptives for the entire dataset (left), and for target dataset with congestive heart failure (CHF) cases (right).

| Variable | Entire Dataset | | | Primary CCS = CHF Only | | |
| --- | --- | --- | --- | --- | --- | --- |
| | N (%) | Mean (SD) | Range | N (%) | Mean (SD) | Range |
| Length of Stay (days) | | 5.92 (±11.42) | 0–3086 | | 5.19 (±5.13) | 0–161 |
| Hospital Mortality (%) | | 3.16 (±17.4) | 0–1 | | 3.17 (±17.5) | 0–1 |
| Primary CCS = 108 * | 25,647 (4.5%) | | | 25,647 (100%) | | |
| Secondary CCS = 158 * | 136,391 (24.1%) | | | 12,385 (48.3%) | | |
| Secondary CCS = 53 * | 241,228 (42.7%) | | | 13,186 (51.4%) | | |
| Secondary CCS = 59 * | 148,528 (26.3%) | | | 9260 (36.1%) | | |
| Secondary CCS = 99 * | 127,175 (22.5%) | | | 11,613 (45.3%) | | |
| Secondary CCS = 101 * | 183,386 (32.5%) | | | 15,053 (58.7%) | | |

* CCS = 108: CHF, CCS = 158: Chronic Kidney Disease, CCS = 53: Lipid Metabolism Disorders, CCS =59: Deficiency/other Anemia, CCS = 99: Hypertension with complications, CCS = 101: Coronary atherosclerosis.

### 3.2. Descriptive Analysis: CHF Comorbidities with the Longest LOS and Highest Hospital Mortality

The average LOS and the mortality rate were calculated for every CHF comorbidity, to find the ones with the highest LOS and hospital mortality rate. As shown in Table 1, CHF patients and a secondary diagnosis of 'gangrene' (CCS = 248), stay at the hospital, on average, 15.70 days. The second, in order, diagnosis associated with a prolonged stay was found to be 'shock' (LOS = 14.76 days), followed by 'intestinal obstruction' (LOS = 13.21 days). Regarding hospital mortality, for CHF patients with a comorbidity of 'cardiac arrest', the mortality rate is 51.7%, followed by 'shock' (mortality rate = 32.9%) and 'septicemia' (mortality rate = 21.9%). Table 2 shows the top ten CHF comorbidities with the longest LOS and the highest rates of hospital death.

**Table 2.** Comorbidities (N > 10) with the longest LOS and the highest mortality rate.

| CHF Comorbidity | LOS (days) | N |
|---|---|---|
| Gangrene (CCS = 248) | 15.70 | 64 |
| Shock (CCS = 249) | 14.76 | 450 |
| Intestinal obstruction w/o hernia (CCS = 145) | 13.21 | 127 |
| Septicemia (CCS = 2) | 13.00 | 456 |
| Acute post-hemorrhagic anemia (CCS = 60) | 12.58 | 356 |
| Aspiration pneumonitis (CCS = 129) | 12.15 | 300 |
| Acute cerebrovascular disease (CCS = 109) | 11.20 | 135 |
| Cardiac arrest and ventricular fibrillation (CCS = 107) | 10.84 | 238 |
| MHSA: Adjustment disorders (CCS = 650) | 10.77 | 53 |
| Complication of surgical /medical procedure (CCS = 238) | 10.49 | 533 |

| CHF Comorbidity | Mortality (%) | N |
|---|---|---|
| Cardiac arrest and ventricular fibrillation (CCS = 107) | 51.7 | 238 |
| Shock (CCS = 249) | 32.9 | 450 |
| Peritonitis and intestinal abscess (CCS = 148) | 26.9 | 26 |
| Septicemia (CCS = 2) | 21.9 | 456 |
| Aspiration pneumonitis (CCS = 129) | 19.3 | 300 |
| Prolapse of female genital organs (CCS = 170) | 18.2 | 11 |
| Intestinal obstruction w/o hernia (CCS = 145) | 18.1 | 127 |
| Liver Ca and intrahepatic bile duct (CCS = 16) | 17.2 | 29 |
| Cancer of the esophagus (CCS = 12) | 15.8 | 38 |
| Gangrene (CCS = 248) | 15.6 | 64 |

### 3.3. Analytical Phase-Task 1: Coefficient Analysis

#### 3.3.1. Length of Stay

To estimate the effect of each individual CHF comorbidity on the hospital LOS, a multiple linear regression model was created, with LOS being the dependent variable. All the CHF comorbidities (dummy CCS variables) were inserted to the model as independent variables. The regression model was found to explain the 30.7% of LOS variability ($R^2 = 0.307$). The regression coefficients provided information about the strength of association between each CHF comorbidity and the LOS. The CHF comorbidity 'gangrene' (CCS = 248) was found to have the strongest effect on the LOS (b = 6.89, $p < 0.001$): When a CHF patient develops 'gangrene' the LOS increases by almost 7 days. Similarly, when a CHF patient develops 'shock', the LOS increases by almost 5 days (b = 4.96, $p < 0.001$). The presence of the CHF comorbidity 'adjustment disorders' increases the LOS by almost 5 days (b = 4.73, $p < 0.001$), while a CHF patient with 'intestinal obstruction without hernia' will have a LOS increase of more than 4 days (b = 4.32, $p < 0.001$). Table 3 presents the top ten CHF comorbidities with the strongest association with the LOS.

**Table 3.** Coefficient analysis of the LOS using multiple linear regression. The table presents the top ten comorbidities associated with the highest LOS increase.

| CHF Comorbidity | b | S.E. | *p*-value |
|---|---|---|---|
| Gangrene (CCS = 248) | 6.89 | 0.55 | <0.001 |
| Shock (CCS = 249) | 4.96 | 0.21 | <0.001 |
| Adjustment disorders (CCS = 650) | 4.73 | 0.59 | <0.001 |
| Intestinal obstruction w/o hernia (CCS = 145) | 4.32 | 0.38 | <0.001 |
| Aspiration pneumonitis (CCS = 129) | 3.63 | 0.25 | <0.001 |
| Acute cerebrovascular disease (CCS = 109) | 3.53 | 0.38 | <0.001 |
| Acute hemorrhage anemia (CCS = 60) | 3.46 | 0.24 | <0.001 |
| Diseases of the mouth (CCS = 137) | 2.89 | 0.61 | <0.001 |
| Complications (surg./med) (CCS = 238) | 2.89 | 0.19 | <0.001 |
| Septicemia (CCS = 2) | 2.71 | 0.21 | <0.001 |

### 3.3.2. Hospital Mortality Rate

To estimate the odds ratio of each individual CHF comorbidity for hospital death (dichotomous outcome), a Multiple Binary Logistic regression model was created. All the CHF comorbidities (dummy CCS variables) were inserted to the model as independent variables. The hospital death indicator was the dependent variable of the model. This regression analysis was conducted to estimate the effect of the CHF comorbidities on the mortality rate, for hospitalized CHF patients. According to findings, 'cardiac arrest and ventricular fibrillation' (CCS = 107) has the strongest association with the mortality rate (OR = 30.50, $p < 0.001$) with odds for hospital death increased by 30 times. The CHF comorbidity 'peritonitis and intestinal abscess' (OR = 14.42, $p < 0.001$) and 'genital organ prolapse' (OR = 12.92, $p < 0.01$) increase the odds for hospital death by 14 and by 13 times respectively. Table 4 shows the top ten comorbidities with the strongest association with hospital mortality.

**Table 4.** Coefficient analysis of mortality rate using binary logistic regression. The table presents the top ten comorbidities associated with the highest odds for hospital death.

| CHF Comorbidity | O.R. | S.E. | *p*-value |
|---|---|---|---|
| Cardiac arrest and ventric. fibril. (CCS = 107) | 30.50 | 0.17 | <0.001 |
| Peritonitis and intestinal abscess (CCS = 148) | 14.42 | 0.63 | <0.001 |
| Prolapse female gen. organs (CCS = 170) | 12.92 | 0.87 | <0.01 |
| Cancer of the esophagus (CCS = 12) | 10.03 | 0.54 | <0.001 |
| Cancer of the liver (CCS = 16) | 8.07 | 0.63 | <0.001 |
| Shock (CCS = 249) | 6.72 | 0.15 | <0.001 |
| Gangrene (CCS = 248) | 4.04 | 0.50 | <0.01 |
| Acute cerebrovascular disease (CCS = 109) | 3.55 | 0.32 | <0.001 |
| Intestinal obstruction w/o hernia (CCS = 145) | 3.15 | 0.32 | <0.001 |
| Respiratory failure; arrest (CCS = 131) | 2.76 | 0.08 | <0.001 |

*3.4. Analytical Phase-Task 2: Dynamic Navigation of CHF Comorbidity Scenarios and Their Effect on Outcomes*

We started by extracting the most frequent CHF comorbidity cluster for a contextually relevant study of the cumulative effect of CHF comorbidities on the LOS and the hospital mortality. To do this, we used the Weka (https://www.cs.waikato.ac.nz/ml/weka/) implementation of k-means, a partitional clustering algorithm. By plotting the within-cluster sum of square errors of the test set for different k's (number of cluster scenarios) we observed error stability (line graph forms an 'elbow') when k = 7. According to the elbow criterion, we proceeded with the parameter k = 7 and generated seven clusters, of which we extracted the most frequent one, for further study. Table 5 shows the output of the simple k-means experiment. Each cluster represents comorbidities that frequently coexist. The cluster that groups most of the instances and that we extracted for further study is Cluster 1. This cluster groups

together the CHF comorbidities: *'Disorders of lipid metabolism', 'deficiency and other anemia', 'hypertension with complications/secondary hypertension', 'coronary atherosclerosis and other heart diseases',* and *'chronic kidney disease'*.

**Table 5.** Clustering of CHF Comorbidities Using the k-means Partitional Algorithm.

| CHF Comorbidities | Clustered Instances |
|---|---|
| **Cluster 1:** 'Disorders of lipid metabolism' (CCS = 53), 'deficiency and other anemia' (CCS = 59), 'hypertension with complications and secondary hypertension' (CCS = 99), 'coronary atherosclerosis and other heart disease' (CCS = 101), 'chronic kidney disease' (CCS = 158) | 7565 (29%) |
| **Cluster 2:** 'Fluid and electrolyte disorders' (CCS = 55), 'nutritional endocrine; and metabolic disorders' (CCS = 58), 'COPD and bronchiectasis' (CCS = 127), 'respiratory failure' (CCS = 131) | 2181 (9%) |
| **Cluster 3:** 'Essential hypertension' (CCS = 98) | 4562 (18%) |
| **Cluster 4:** 'Essential hypertension' (CCS = 98), 'disorders of lipid metabolism' (CCS = 53), 'coronary atherosclerosis and other heart disease' (CCS = 101), 'cardiac dysrhythmias' (CCS = 106) | 5759 (22%) |
| **Cluster 5:** 'Cardiac dysrhythmias' (CCS = 106), 'fluid and electrolyte disorders' (CCS = 55), 'deficiency and other anemia' (CCS = 59), 'hypertension with complications/secondary hypertension' (CCS = 99), 'chronic kidney disease' (CCS = 158), 'heart valve disorders' (CCS = 96), 'pulmonary heart disease' (CCS = 103) | 2098 (8%) |
| **Cluster 6:** 'Deficiency and other anemia' (CCS = 59), 'hypertension with complications and secondary hypertension' (CCS = 99), 'chronic kidney disease' (CCS = 158), 'coronary atherosclerosis and other heart disease' (CCS = 101), 'COPD and bronchiectasis' (CCS = 127), 'respiratory failure; insufficiency; arrest (adult)' (CCS = 131), 'diabetes mellitus without complications' (CCS = 49), 'acute and unspecified renal failure' (CCS = 157) | 2284 (9%) |
| **Cluster 7:** 'Respiratory failure; arrest' (CCS = 131), 'cardiac dysrhythmias' (CCS = 106), 'fluid and electrolyte disorders' (CCS = 55), 'essential hypertension' (CCS = 98), 'screening and history of mental health and substance abuse' (CCS = 663) | 1198 (5%) |

The next step involves the estimation of the mean LOS and mortality rate for every different combination inside Cluster 1 (CCS = 53, CCS = 59, CCS = 99, CCS = 101, CCS = 158). For every combination, new dummy variables were stored into the database to facilitate the creation of conditional probability tables, on-demand (Table 5). For instance, for the combination 'lipid metabolism disorders' (CCS = 53) and 'deficiency and other anemia' (CCS = 59), a new variable will be generated, based on the condition:

IF CCS(53) = 1 AND CCS(59) = 1 AND CCS(99) = 0 AND CCS(101) = 0 AND CCS(158) = 0
THEN [CCS(53) Λ CCS(59)] = 1
ELSE [CCS(53) Λ CCS(59)] = 0

This condition triggers the addition, to the dataset, of a new variable, with values of '1' for instances where {CCS53 = 1, CCS59 = 1, CCS99 = 0, CCS101 = 0, CCS158 = 0}. We then calculated, the mean LOS, the mortality rate, and the 95% C.I of the means, for the '1' cases, for all possible combinations of the Cluster 1 contents (total = $2^5$ = 32 combinations) and constructed conditional probability tables, that show the cumulative effect of any comorbidity construct. The conditional probability tables were also visualized with step-by-step Bayesian graphs of comorbidity constructs, to better understand the additive effect of CHF comorbidities on the outcomes under study. Table 6 shows the mortality rate, mean length of stay and confidence intervals for each one of the 32 different combinations from the cluster. The combinations are organized as cumulative paths of comorbidity constructs.

**Table 6.** Mortality rate and mean LOS for every CHF Comorbidity combination within Cluster 1.

| Different Paths of CHF Comorbidities | N | Mortality rate (%) (95% C.I) | Mean LOS (days) (95% C.I) |
|---|---|---|---|
| No comorbidity from cluster | 2800 | 3.68 (2.98–4.37) | 4.76 (4.55–4.97) |
| 53 | 1598 | 2.37 (1.63–3.12) | 4.26 (4.10–4.43) |
| 53+59 | 508 | 1.57 (0.49–2.65) | 5.17 (4.83–5.51) |
| 53+99 | 95 | 0.00 (0.00–0.00) | 4.63 (3.76–5.49) |
| 53+101 | 3008 | 2.09 (1.58–2.60) | 4.32 (4.20–4.44) |
| 53+158 | 79 | 3.79 (0.00–8.01) | 5.11 (4.25–5.98) |
| 53+59+99 | 21 | 4.76 (0.00–14.09) | 5.23 (3.61–6.86) |
| 53+59+101 | 995 | 2.21 (1.34–3.08) | 5.29 (4.98–5.60) |
| 53+59+158 | 56 | 7.14 (1.24–13.04) | 6.80 (5.56–8.04) |
| 53+99+101 | 144 | 1.38 (0.00–3.31) | 5.50 (4.41–6.59) |
| 53+99+158 | 902 | 2.54 (1.56–3.53) | 5.08 (4.79–5.36) |
| 53+101+158 | 220 | 4.55 (1.79–7.30) | 5.09 (4.57–5.62) |
| 53+59+99+101 | 51 | 3.92 (0.00–9.30) | 6.27 (4.53–8.01) |
| 53+59+99+158 | 920 | 2.93 (1.88–3.98) | 6.04 (5.72–6.35) |
| 53+59+101+158 | 133 | 6.01 (1.98–10.05) | 6.16 (5.19–7.13) |
| 53+99+101+158 | 2308 | 3.25 (2.52–3.97) | 4.93 (4.74–5.12) |
| 53+59+99+101+158 | 2148 | 2.61 (1.93–3.28) | 5.82 (5.60–6.04) |
| 59 | 898 | 4.34 (0.68–3.01) | 5.54 (5.25–5.82) |
| 59+99 | 42 | 2.38 (0.00–7.04) | 6.71 (4.94–8.48) |
| 59+101 | 684 | 4.38 (2.88–5.89) | 5.57 (5.22–5.93) |
| 59+158 | 170 | 4.70 (1.52–7.89) | 6.06 (5.36–6.77) |
| 59+99+101 | 34 | 2.94 (0.00–8.79) | 6.18 (4.26–8.01) |
| 59+99+158 | 1088 | 3.03 (2.04–4.02) | 6.10 (5.74–6.45) |
| 59+101+158 | 218 | 6.42 (3.17–9.67) | 6.18 (5.57–6.78) |
| 59+99+101+158 | 1294 | 2.71 (1.82–3.58) | 6.11 (5.77–6.45) |
| 99 | 127 | 3.15 (0.11–6.19) | 4.96 (4.07–5.85) |
| 99+101 | 76 | 3.94 (0.00–8.32) | 5.55 (4.28–6.82) |
| 99+158 | 1046 | 4.68 (3.43–5.94) | 5.62 (5.16–6.08) |
| 99+101+158 | 1317 | 4.25 (3.17–5.33) | 5.43 (5.17–5.69) |
| 101 | 2181 | 2.75 (2.06–3.44) | 4.80 (4.59–5.01) |
| 101+158 | 242 | 6.20 (3.16–9.23) | 6.28 (5.48–7.08) |
| 158 | 244 | 7.79 (4.42–11.15) | 5.88 (4.84–6.93) |

### 3.4.1. Directed Acyclic Graphs for Comorbidity Construct Scenarios

After having estimated the LOS and mortality rate for the different combination constructs of the cluster {*metabolism disorders, anemia, hypertension with complications, coronary atherosclerosis, chronic kidney disease*}, results were visualized with directed acyclic graphs. Each new graph node represents an addition of a new diagnosis on top of the preceding one. The user can follow any path, as shown in Figure 2 and see the updated LOS and mortality rate. In the majority of the paths, the mortality rate and the LOS increases as more CHF comorbidities are added. Characteristically, for CHF patients who only have 'disorders of lipid metabolism' (CCS = 53), the mean LOS is 4.26 days (95% C.I = 4.10–4.43). When 'deficiency and other anemia' (CCS = 59) is added to the profile, the mean LOS increases to 5.17 days (95% C.I = 4.83–5.51). When on top of these two comorbidities, the patient is diagnosed with 'hypertension with complications' (CCS = 99), the mean LOS further increases to 5.23 days (95% C.I = 3.61–6.86). Finally, a new LOS increase is observed with the addition of 'coronary atherosclerosis' (CCS = 101), up to 6.27 days (95% C.I = 4.53–8.01). In the same manner, while the exclusive presence of 'coronary atherosclerosis and other heart diseases' (CCS = 101) is associated with a mean mortality rate of 2.75%, when 'chronic kidney disease' (CCS = 158) is added to this patient scenario, the mortality rate increases up to 6.19%.

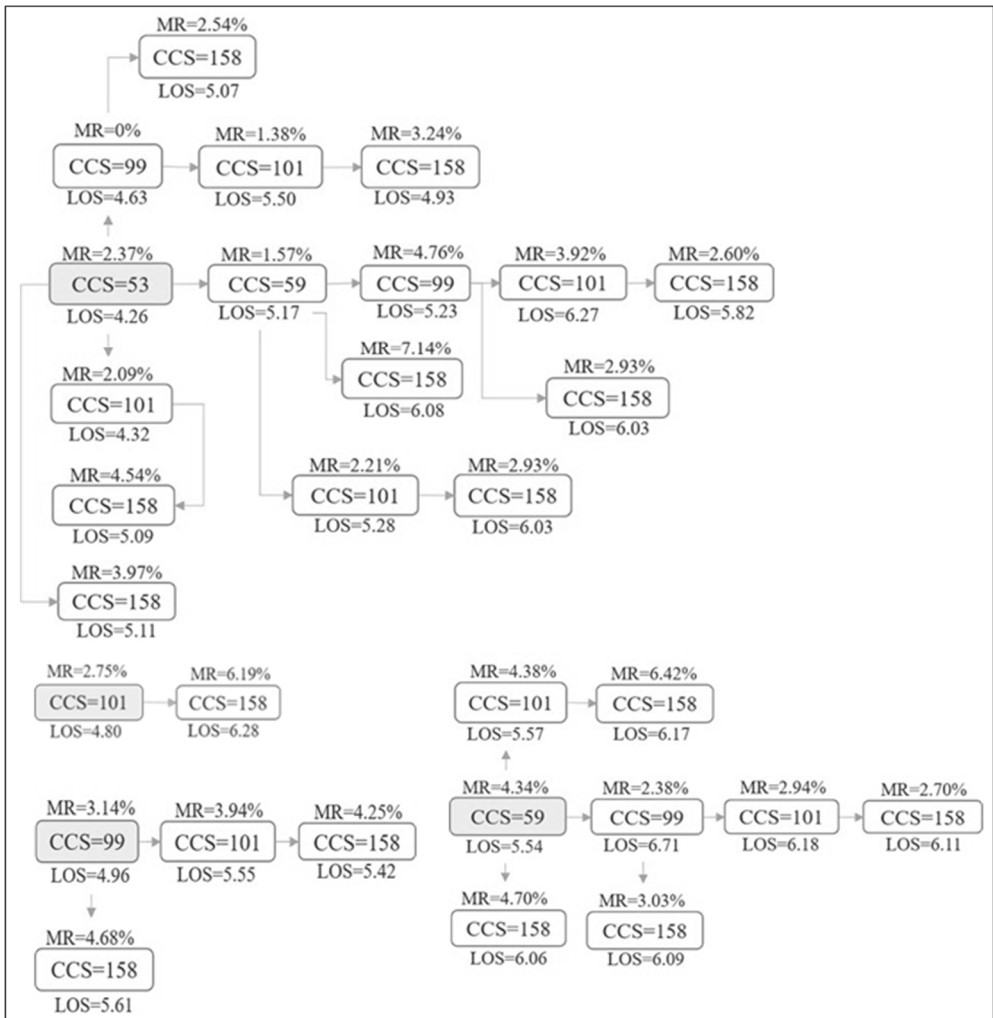

**Figure 2.** Effects of comorbidity constructs on LOS and mortality rate.

Interestingly, some comorbidities may have a different effect on outcomes, depending on the pre-existing disease set that the new diagnosis has been added onto. For instance, although the mean LOS and the mortality rate both remain the same when 'coronary atherosclerosis' (CCS = 101) is added on top of 'disorders of lipid metabolism' (CCS = 53), this is not the case when CCS = 101 is added on top of CCS = 99 (hypertension with complications). Evidently, for patients with CHF, the 'coronary atherosclerosis' and 'hypertension with complications' combination is a more severe 'diagnosis blend' than that of 'coronary atherosclerosis' with 'disorders of lipid metabolism'. As Table 6 shows, the bigger the number of coexisting CHF comorbidities, the longer the LOS. This is not the case, though, for the mortality rate, as shown in Table 7. Apparently, there may exist other comorbidities, outside the cluster under investigation with a stronger association with the risk for hospital death.

**Table 7.** Mortality rate and mean LOS according to the number of comorbidities present.

| Total Comorbidities Present | N | Mortality Rate (%) (95% C.I) | Mean LOS (days) (95% C.I) |
|---|---|---|---|
| No comorbidity from cluster | 2800 | 3.68 (2.98–4.37) | 4.76 (4.55–4.97) |
| 1 out of 5 comorbidities | 5048 | 3.17 (2.69–3.65) | 4.82 (4.68–4.95) |
| 2 out of 5 comorbidities | 5950 | 3.03 (2.59–3.46) | 4.94 (4.82–5.07) |
| 3 out of 5 comorbidities | 4995 | 3.32 (2.84–3.81) | 5.52 (5.38–5.66) |
| 4 out of 5 comorbidities | 4706 | 3.12 (2.63–3.61) | 5.52 (5.37–5.67) |
| All five comorbidities | 2148 | 2.61 (1.93–3.28) | 5.82 (5.60–6.04) |

### 3.4.2. Bayesian Networks of CHF Comorbidities

The second goal of task 2 was to construct Bayes Networks for CHF comorbidities included in Cluster 1, for the dichotomous outcome 'hospital death'. While we previously visualized all combinations of comorbidity constructs, we now present a supervised representation of associations between comorbidities and the 'hospital death', based on a trained Bayes Net model. Using Weka, we developed a Bayes Network using the Simple Estimator, which estimates the conditional probability tables of a Bayes network once the structure has been learned. We used a Genetic Search algorithm with Markov blanket correction. The Markov blanket correction was applied to the network structure, to enhance the information included in the values of the parents and children of the network nodes. The model was trained using the original dataset and was tested using a subset from a previous CMS reporting year. Apparently, the construction of the Bayes Network in this study is navigational and is not meant to be used for prediction purposes. It is, therefore, intended to be used as a 'navigator' of comorbidity associations.

The top patent node of the trained Bayes Network is CCS=158 (chronic kidney disease). Its four child notes are the four remaining comorbidities of Cluster 1 (most frequent CHF comorbidity group). The outcome under investigation (hospital death) has one direct parent node: CCS = 53 (lipid metabolism disorders). Figure 3 shows the network, as it was visualized by the Weka Classifier Graph Visualizer (right) and the probability distribution tables to the 'hospital death' node for the path that leads to the outcome of interest. This path is {(CCS158→CCS101)→CCS53→DEATH}. While we hereby present the Bayes Network for Cluster 1, similar Bayes Network graphs can be generated, following the same approach, for any of the remaining CHF comorbidity clusters.

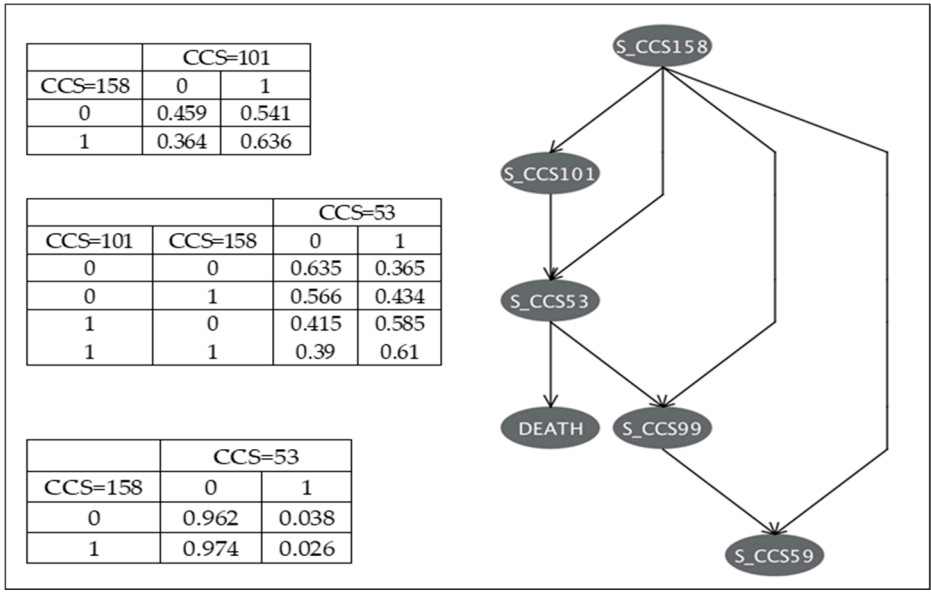

**Figure 3. Right:** Bayesian Network of CHF Comorbidities included in the most frequent cluster. **Left:** Probability distribution tables for the path {(CCS158→CCS101)→CCS53→DEATH}.

## 4. Discussion

This study examined the prevalence and cumulative effect of CHF comorbidities for elderly patients with a primary diagnosis of CHF, using medical claims data. The study presented an in-depth exploration of comorbidities for one of the most frequent reasons for hospitalization in the United States and also introduced Bayesian methods to understand the cumulative burden of comorbidities on negative outcomes of hospital care.

A significant burden of comorbidity was observed in CHF patients with two or more comorbidities. The most common CHF comorbidities associated with an increased LOS were found to be 'gangrene',

'shock', 'intestinal obstruction without hernia', and 'adjustment disorders'. Another study that examined psychosocial factors and other comorbidities in CHF patients, also found that 'adjustment disorders' are associated with as increasing LOS and mortality rate [22,31]. The regression-based coefficient analysis showed that for an increase of 1 unit to value of the 'gangrene' variable (change from '0' to '1'), the LOS increases by 6.90 days (p < 0.001). Similarly, other secondary $D_x$'s that were found to be associated with an increased LOS were found to be: 'Shock' (LOS = 4.97, p < 0.001) [32], 'adjustment disorders' (LOS = 4.73, p < 0.001), and 'intestinal obstruction without hernia' (LOS = 4.33, p < 0.001). 'Anemia' in patients with CHF was also found to be associated with prolonged hospital stay in a previous study [33], with findings similar to our current study.

As far as the mortality rate is concerned, we found that comorbidities associated with increased mortality rate are the 'cardiac arrest and ventricular fibrillation', 'acute cerebrovascular disease' [34], 'peritonitis and intestinal abscess', 'prolapse of female genital organs', 'shock', and 'cancer of the esophagus'. Results of multiple binary logistic regression revealed that the CHF comorbidity 'cardiac arrest and ventricular fibrillation' (CCS = 107) is associated with a 30.5 increase to the odds for hospital death. Other CHF comorbidities that were found to increase mortality risk are 'peritonitis and intestinal abscess' (OR = 14.4), 'prolapse of female genital organs' (OR = 12.9), 'cancer of the esophagus' (OR = 10.0), 'shock' (OR = 6.721) [35], and 'acute cerebrovascular disease' (OR = 3.559) [36]. In similar studies, the 'acute myocardial infarction', and 'acute and unspecified renal failure' diagnoses were also found to be associated with a significant increase to the odds for hospital death, for patients with a primary diagnosis of CHF [37,38].

Several studies have shown that comorbidities have a significant impact on survival and LOS in CHF patients, and our study results are in agreement. Our study shows that comorbidities can have a variable effect of these outcomes, according to the comorbidity construct they belong to. We, therefore, recognize the need for the development of comorbidity-specific software risk estimation add-ons to existing clinical decision support systems that quantify the different risk levels for those patients. This will facilitate data-driven, informed decision making and improved patient counseling. The need for such systems and mechanisms have been discussed and recommended in the literature [34] in an effort to assist physicians to provide "individualized person-centered care" [31].

Our study and the "block-by-block" comorbidity construction approach is an effort to this direction. It is imperative for physicians to recognize common comorbidities for their patients and understand the effect of comorbidities on outcomes of care. As this work shows, for CHF patients, different comorbidity constructs may have a variable effect on the outcomes. We recognize significant clinical implications associated with our findings. Firstly, by identifying the prevalence and quantifying their cumulative effect for patients, clinical decision-makers will have, at their disposal, evidence for informed clinical decision making in an ongoing effort for improvements to the quality of care. In addition, since many patients oftentimes develop hospital-acquired conditions, which may or may not be associated with the primary diagnosis, it is important for physicians to learn, on-the-fly, how these newly diagnosed conditions increase the risk for negative outcomes. This can be the foundations for proactive clinical decisions, in anticipation of high risk for negative clinical events, such as complications or death. The authors recognize as a priority for health systems to explore opportunities for the development of comorbidity-specific recommender tools that would be used by clinical decision-makers and quality improvement specialists. The authors finally believe that education and training of medical professionals and residents should utilize large healthcare datasets, and assist future professionals in recognizing common comorbidities, and their effect on critical outcomes of care.

A limitation of this study is that it was conducted with a five-year-old medical claims data file. The year 2014 was the last year where ICD-9-CM was used for $D_x$ coding. While the 2015 dataset was the most recent dataset available to the researchers, it includes mixed ICD-9 and ICD-10 coding (since ICD-10 rolled out in October 2015). For this reason, it was not used in this study to avoid partial data removal for the fall season months. Although disease patterns do not change over the course of five

years, we do believe that it would be beneficial for the research to have been conducted with the use of a more recent dataset. Finally, while we found interesting LOS and mortality trends during the construction of the CHF comorbidity sets, the study did not control for any comorbidities other than {*CCS = 53, CCS = 59, CCS = 99, CCS = 101, CCS = 158*}, which could have a strong association with the outcomes under study.

**Author Contributions:** Conceptualization, D.Z.; methodology, D.Z.; validation, D.Z., N.R., and S.Z.; formal analysis, D.Z, and S.Z.; resources, D.Z.; data curation, D.Z.; writing—original draft preparation, D.Z.; writing—review and editing, D.Z. and N.R; visualization, D.Z.; supervision, D.Z.; project administration, D.Z.; funding acquisition, D.Z.

**Funding:** This research was funded by the CENTRAL MICHIGAN UNIVERSITY FACULTY RESEARCH AND CREATIVE ENDEAVORS (FRCE) GRANT, grant number 4830, 01/01/2019 – 06/30/2020, P.I: Dimitrios Zikos.

**Conflicts of Interest:** The authors declare no conflict of interest.

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
