# Peer review of "A Bayesian Study of the Dynamic Effect of Comorbidities on Hospital Outcomes of Care for Congestive Heart Failure Patients"

_technologies, doi:10.3390/technologies7030066_

Round 1
Reviewer 1 Report
In this version, the discussion contains a more complete evaluation of the clinical implications of the presented research. The paper can be published.
Reviewer 2 Report
The authors have addressed my previous concerns. However, please improve the quality of their figures, especially Fig. 1 I can't see anything. After that, I think it can be accepted for publication.
This manuscript is a resubmission of an earlier submission. The following is a list of the peer review reports and author responses from that submission.
Round 1
Reviewer 1 Report
The paper presents a statistical based study of the relations between co-morbidities and some parameters that affect the quality of life and the cost of care.
The paper is well written, the used methods are scientifically based, and the results are widely explained. The authors should explain in a clearer way the applications and the consequences of their results in the hospital routine life.
Author Response
Thank you for your recommendation to discuss clinical implications for the daily routine life. We have improved portion of the discussion section and added new material in order to explain how findings may improve clinical practice. Below is the paragraph covering the aforementioned discussions:
“We recognize significant clinical implications associated with our findings. Firstly, by identifying the prevalence and quantifying their cumulative effect for patients clinical decision-makers will have, at their disposal, evidence for informed clinical decision making in an ongoing effort for improvements to the quality of care. In addition, since many patients oftentimes develop hospital-acquired conditions, which may or may not be associated with the primary diagnosis, it is important for physicians to learn, on-the-fly, how these newly diagnosed conditions increase the risk for negative outcomes. This can be the foundations for proactive clinical decisions, in anticipation of high risk for negative clinical events such as complications or death. There is more research to be done in order to develop and provide comorbidity-specific recommender tools to clinical decision-makers and quality improvement specialists. The authors finally believe that education and training of medical professionals and residents should utilize large healthcare datasets, and assist future professionals in recognizing common comorbidities, and their effect on critical outcomes of care.”

Reviewer 2 Report
There are some grammatical errors and typos in this manuscript. The authors have to re-check and revise carefully. For example: "... there an anticipated ongoing increase ...", "... can contribute to improving ...", etc. Section "Materials and Methods" has not been written clearly. There is a need for providing more description on the method such as on regression algorithm. CMS provided a big number of dataset, why did the authors only select 5% to perform experiments? If they can do experiments with big data, the results may be better and reliable. Can the authors test their method on the latest dataset, e.g., data of 2017/2018? There is a need for providing more discussions on descriptive analysis, such as to answer why Gangrene was high in LOS, but was very low in Mortality. The authors should mention that Weka has been applied to a lot of works in biomedical fields, such as https://doi.org/10.1016/j.ab.2018.06.011, https://doi.org/10.1002/jcc.24842, and https://doi.org/10.1186/s12859-016-1369-y. There are totally 32 combinations, but why did the authors only show 13 in Table 5? The quality of figures has to be improved. Figure 3 needs to be explained more. What are the limitation and disadvantage of this work? In abstract, CMS was used without abbreviation definitionAuthor Response
Please see the attachment

Round 2
Reviewer 1 Report
Also after the revision, I think I can maintain, my previous referral on the manuscript.
The paper is well written, the used methods are scientifically based, and the results are widely explained.
The authors should explain in a clearer way the applications and the consequences of their results in the hospital routine life. I can see only a small improvement on this point in the revised version of the manuscript (in the discussion section). They should indicate some examples of this possible applications.
Reviewer 2 Report
The authors did not address most of my previous concerns. Mostly, they inserted some discussions to explain why they did not address them, but I think it is not enough to gain the study's quality.
For example, the authors only used 5% of CMS data to perform experiments, what are the range of them and did the authors convince that 5% will help them produce a good result. At least, the authors should try 2 following experiments: performing their method on a bigger dataset, testing their method on a different dataset (not CMS). It will help this study gain quality.
Section "Methods" has not been improved as I suggested from the previous concerns. Now it is still not clear. Moreover, all the methods used in this study are well-established methods. What are the innovations of this work?
The quality of figures has not been improved yet.